# Depressive and Anxiety Disorders of Parents of Children with Cancer

**DOI:** 10.3390/jcm11195670

**Published:** 2022-09-26

**Authors:** Eleonora Mess, Weronika Misiąg, Tomasz Klaszczyk, Kamila Kryś

**Affiliations:** 1Department of Clinical Nursing, Division of Oncology and Palliative Care, Faculty of Health Science, Wroclaw Medical University, 50-367 Wroclaw, Poland; 2Faculty of Medicine, Wroclaw Medical University, 50-367 Wroclaw, Poland; 3Department of Psychology, University of Warsaw, 00-183 Warszawa, Poland; 4Department of Children’s Immunology, J. Gromkowski Regional Specialist Hospital, 5 Koszarowa Street, 51-149 Wroclaw, Poland

**Keywords:** depression, anxiety, QoL, childhood cancer, parents, chemotherapy complications

## Abstract

Every year in Poland there are approximately 1200 new cases of malignant tumours in children and adolescents. Leukaemia, CNS tumours, and lymphomas are the most frequently diagnosed cancers. Coping with a child’s illness is challenging, which is why many parents suffer from anxiety, depression disorders, and even PTSD (post-traumatic stress disorder). The aim of this study is to assess the anxiety and depression levels of carers of children with cancer. Method: The study participants were 101 carers of children suffering from cancer. The study was conducted using standardized questionnaires: the Zung ccale, HADS scale, and Karnofsky scale, and three questionnaires designed by the author. Results: According to the results of the Karnofsky scale, carers’ anxiety and depression levels were negatively affected by their children’s poor performance. The younger age of children significantly correlated with the severity of depression in their parents. HADS and Zung scale anxiety levels were observed to have statistically significant effect on the severity of depressive disorders. Conclusions: Receiving a diagnosis of childhood cancer contributes to the incidence of depression and anxiety disorders among carers. Carers’ anxiety and depression levels were strongly associated with their child’s age and their child’s performance.

## 1. Introduction

Every year in Poland, approximately 1200 new cases of malignant tumours are diagnosed in children and adolescents aged 0 to 17 years [1]. Polish National Cancer Registry data show that 187 children aged 0 to 19 years died from cancer in 2019. Leukaemia, CNS tumours, lymphomas, and nephroblastoma are the most frequently diagnosed types of cancer in Poland and internationally [2,3]. Worldwide nearly 400,000 children develop cancer every year. Globally, according to data published by WHO, acute lymphoblastic leukaemia (ALL) is the most common childhood cancer.

It is estimated that ALL accounts for 19% of total childhood cancer, while non-Hodgkin lymphoma accounts for 5%, nephroblastoma 5%, Burkitt lymphoma 5% and retinoblastoma 5% [4,5]. Despite developments in the field of diagnostics and a range of treatment options, as well as increasingly better survival statistics (70–80% cure rate) [6], the moment a child is diagnosed with cancer is extremely stressful for parents [7]. The diagnosis of a life-threatening illness is associated with increased anxiety and reorganization of the whole family’s life. The nature of the disease, frequent hospitalizations [8], prolonged treatment involving numerous complications, fear of death or relapse [9,10,11,12], and constant stress undeniably affect the quality of life of the entire family. The new situation forces parents to go through different stages of adjustment to the illness. In such circumstances, any model of adaptation will influence the optimal functioning of the child.

Coping with a child’s illness is challenging, so many parents exhibit anxiety and depressive disorders [13,14,15], and even post-traumatic stress disorder (PTSD) [16,17]. Parents’ anguish and stress may contribute to increased worry in the child [18]. It is extremely important to understand the different stages of treatment and the complexity of the whole situation, as well as to provide psychological care for the whole family, as the suffering of the parent has a significant impact on the child’s adaptation to the illness, their recovery, and functioning [13,14].

The aim of this study is to assess levels of anxiety and depression among carers of children with cancer, and to examine the influence of the child’s age, the number of complications connected with chemotherapy, and the performance of the child on the severity of parents’ mental health disorders.

## 2. Materials and Methods

One hundred and one carers of children suffering from cancer participated in the qualitative-quantitative study. The surveys were conducted between January 2018 and March 2018 at the Department of Bone Marrow Transplantation, Oncology and Paediatric Haematology in Wroclaw. The observational study was conducted using standardized questionnaires: the Zung Self-Rating Depression Scale (ZSDS), the Hospital Anxiety and Depression Scale (HADS), the Karnofsky Performance Scale, and three anonymous questionnaires designed by the authors, which included socio-demographic assessment as well as assessment of chemotherapy complications and their severity. The designed questionnaires are included in the Appendix A. 

The aim of the research was to assess whether a child’s cancer affects the occurrence of depressive and anxiety disorders in their parents. A further purpose of the study was to identify which factors influence the development of mental disorders.

The study group was informed about the purpose of the research and the possibility of withdrawing from participation in the study. Approval No. KB-16/2018 to conduct the study was obtained from the Ethics Committee.

The Zung Self-Rating Depression Scale assesses the severity of depression symptoms. The questionnaire consists of 20 questions and the patient indicates on a 4-point scale the answer that best represents their current well-being. The higher the score obtained, the more severe the depression symptoms. The score is quantified and falls into four ranges that identify the severity of depression. Scores below 50 indicate no depression, 50–59—mild depression, 60–69—moderate depression, and 70–80—severe depression [19].

The Hospital Anxiety and Depression Scale (HADS) consists of 16 questions with four possible answers, the results of which are interpreted as follows: 0–7 points means no depressive disorder, 8–10—borderline depressive symptoms, 11–12—symptoms of significant severity. The anxiety component is assessed similarly [20,21].

The Karnofsky Performance Scale assesses the patient’s performance and level of functioning in daily activities. This questionnaire assesses the level of independence and self-sufficiency in meeting personal needs, as well as the degree of dependence on institutional medical care. This questionnaire returns a score from 0 to 100 points. On this scale, 11 categories of results are distinguished. A score of 0 means death, 10—a sudden increase in the threat to life, 20—very sick patient with active supportive treatment, 30—severely disabled, 40—disabled, 50—requires frequent medical care, 60—requires occasional assistance, 70—unable to carry out normal activity but cares for self, 80—normal activity with effort and some signs or symptoms of disease, 90—able to continue normal activities with minor signs of disease. The highest score (100 points) indicates absence of complaints and symptoms and a good quality of life [22]. This scale was used to assess the level of independence of the affected child and to consider the effect of that score on depression in parents.

The authors’ own questionnaires included questions about the social activities of carers of children with cancer, since diagnosis. The questionnaire on complications in children after chemotherapy included questions about the child’s age and gender, the type of cancer diagnosed, the duration of the disease, and the type of complications that occurred after treatment with cytostatic agents. Severity was assessed using a 10-point visual scale, where 0 meant that the symptom was not present and 10 that the symptom in question significantly impeded daily functioning.

## 3. Statistics

The mean, standard deviation, median, first and third quartiles, minimum and maximum, and interquartile range (IQR) were employed to analyse quantitative variables, while the number and percentage of occurrences of each value were considered in order to assess qualitative variables. Using the chi-squared test (with Yates correction for 2 × 2 tables) and Fisher’s exact test for samples with low expected values, the values of qualitative variables were compared across groups. In cases where normal distribution was not obtained, analysis of variance (ANOVA) (normal distribution) and Kruskal–Wallis testing were applied to compare the values of quantitative variables.

To identify statistical significances between different groups, post-hoc analysis was conducted using Fisher’s LSD test (for normal distribution) or, in other cases, Dunn’s test. Pearson’s and Spearman’s correlation coefficients were applied to analyse the correlation between quantitative variables, where |r| ≥ 0.9 was interpreted as a very strong correlation and |r| < 0.3 as a negligible correlation. The normality of the variables’ distribution was verified using the Shapiro–Wilk test. The analysis was conducted using the R programme, version 3.4.3 (R Foundation, Vienna, Austria), and *p*-values ≤ 0.05 were regarded as statistically significant.

## 4. Results

### 4.1. General Statistics of the Children

The results of the questionnaires showed that among the children there were 53 (52.48%) boys and 47 (46.53%) girls, while one person did not answer this question. The mean age of the children was 8.87 years (SD = 4.9) (Table 1). Among the respondents, leukaemia was found to be the predominant disease entity (58 patients), followed by nervous system tumours (18 patients), sarcomas (9 patients), and lymphomas (5 patients), while 10 of the patients were classified as having “other types of cancer” (Table 1). The mean age of the children at diagnosis was 7.71 years (SD = 5.18). The youngest child diagnosed with cancer was 5 months old, and the oldest was 17 years and 5 months old. The largest group among the respondents were children whose illness lasted for up to 6 months—33.66%. Disease duration of 7 to 12 months was observed in 32.67% of the children, 13 to 24 months in 11.88%, and 20.79% of the children were ill for more than 24 months (Table 1).

### 4.2. Zung and HADS Scale

The results of the Zung Self-Rating Depression Scale showed no depression symptoms in 90 respondents, mild depression in 10, and moderate depression in one parent. No signs of severe depression were observed in any of the study participants (Table 2). Referring to the results involving the HADS anxiety scores, 42 out of 101 parents experienced marked anxiety disorders, 23 respondents had a borderline condition, while 36 experienced no anxiety disorders. According to the HADS depression scores, 52 respondents had no depressive disorders, 26 had borderline condition, and 23 had a score indicating marked depressive disorders (Table 3).

Using Spearman’s correlation coefficient, a statistically significant association was found (*p* < 0.05) between the level of anxiety according to the HADS and Zung scores and the severity of depressive disorders (Table 4). The correlation was positive; the higher the level of anxiety, the higher the severity of depressive disorders.

### 4.3. Activity Level of the Child

On the Karnofsky scale assessing patients’ performance, the mean number of points obtained by the respondents was 68.8 (SD = 20.71) out of a possible 100. It can therefore be concluded that, on average, the patients’ condition was defined as “a state of being unable to perform work or normal activity while retaining the ability to perform daily activities” (Table 5).

Examining the effect of a child’s performance according to the Karnofsky scale on their carers’ anxiety and depression levels, a significant statistical relationship was demonstrated (*p* < 0.05). An association was observed indicating that the better the child’s performance, the lower the severity of the parents’ mental disorders (Table 6).

### 4.4. The Daily Activities of Caretakers

A study of the daily activities of carers of children with cancer returned results in which 87.13% of respondents stated that they left the house less often for leisure activities, 86.14% of parents reported less frequent meetings with family and friends, 78.22% admitted devoting less time to their hobbies and interests, and 80.2% worked fewer hours or had to stop working because of their child’s illness. Some respondents experienced limited contact with people, less interest in other people’s problems, irritability, concentration problems, reduced effectiveness at work, or deterioration of relationships with loved ones, while 48.51% of the study participants continued to enjoy an active life. The results of the survey are presented in Table 7.

### 4.5. Complications after Treatment

Complications after chemotherapy occurred in 96 children, two children had none, while three respondents did not answer this question. The most commonly reported complications included decreased appetite (88.12%), weakness (82.18%), hair loss (77.23%), nausea and vomiting (76.24%), and pale skin (66.34%). These and other complications are presented in Table 8. The mean overall severity of complications was 3.19 points on a 10-point scale (SD = 1.43). Decreased appetite, weakness, nausea, and vomiting were the complications with the highest levels of severity in affected children. The least severe complications included dizziness, headaches, and constipation (Table 9).

### 4.6. Correlations

No correlation was observed between cancer type and the occurrence of complications after chemotherapy (*p* > 0.05). Similarly, the overall severity of complications was not dependent on the type of cancer.

According to the Zung scale, we found that the severity of depressive disorders was strongly associated with child’s age (*p* < 0.05). The older the child, the lower the severity of the disorder (Table 10).

No effect on parents’ anxiety and depression levels was observed for the number of complications after chemotherapy (*p* > 0.05). The analysis of the correlation coefficients between the duration of the child’s illness and parents’ anxiety and depression levels showed no statistically significant results. The severity of anxiety and depressive disorders was not found to depend significantly on the duration of the disease (Table 11).

## 5. Discussion

Cancer can affect children at any age. Each year, more than 400,000 new cases of childhood cancer are diagnosed worldwide in children aged 0 to 19 years [5]. Despite great advances in the field of diagnostic methods, as well as increasingly better treatment results and improved survival statistics, a child’s cancer represents a crisis for the whole family [6,7].

### 5.1. Influence of the New Diagnosis on the Functioning of the Family

It is devastating to learn that a child has cancer. The fear of a previously unknown problem arises, there is a need to take on a new role, and uncertainty about covering the cost of the child’s treatment or the possible relocation of the family nearer to the treatment centre [23,24]. The daily life of the whole family is disrupted, and their quality of life significantly decreases. A child’s chronic illness negatively affects daily responsibilities, work life, family, and social relationships [13,25,26,27,28]. Receiving news of a life-threatening diagnosis is a difficult emotional experience for parents. Shock, panic, and disbelief predominate after receiving the initial diagnosis and during the initial treatment, with some parents experiencing denial, anger, sadness, and guilt [29,30,31].

### 5.2. Coping Strategies

Most often, parents’ responses to the news of a child’s cancer can be divided into several stages: stage I—denial and isolation; stage II—anger and opposition; stage III—bargaining; stage IV—depression; and stage V—coming to terms with the situation, hope, faith, and trust. These stages are consistent with the stages of grief described by Elisabeth Kübler-Ross and discussed in her book titled On Death and Dying [32]. Prolonged negative emotional responses influence the development of an increased sense of hopelessness [25,33,34], and the occurrence of anxiety disorders, depressive disorders, and even PTSD [35,36,37].

### 5.3. Mental State

The predominant depressive symptoms of carers of children undergoing cancer treatment are helplessness, a sense of hopelessness, sadness, guilt, decreased energy, anxiety, and problems with sleep, concentration, and decision-making [25,38,39,40,41,42]. Parents’ mental states can make it difficult for them to properly take care of the sick child who requires intensive treatment and psychosocial care. Parents who are at increased risk of developing depressive disorders require thorough and timely identification, and should be provided with professional multidisciplinary support to improve their mental health. Studies have reported higher levels of stress, depression, anxiety, and hopelessness in parents of children with cancer, compared with parents of healthy children or children with other chronic illnesses or disabilities [13,43,44]. Other studies show that family members can be described as cosufferers from cancer, and that the majority of them have depression [45,46,47,48].

### 5.4. Quality of Life

Cancer brings considerable demands on family members and can have a great impact on their psychological health. Quality of life among family members of cancer patients is generally poor and is a risk factor for other disorders [49,50,51,52,53,54]. Their life satisfaction is also significantly lower than other groups [55].

### 5.5. The Parent’s Gender and Impact on Marriage

The results of the study by Iqbal et al. showed a higher prevalence of depression among mothers of children with cancer [23,24,25,40,56,57]. Studies show that fathers and mothers both reported greater marital distress [56]. Higher levels of anxiety and depression have an impact on marital dissatisfaction, and affected couples are more likely to report unhappiness in their marriage [58,59].

### 5.6. Socioeconomic Factors

It has also been noted that lower socioeconomic status, low financial income, and low education levels contribute to the severity of parents’ depression [13,25,35]. The most commonly reported concerns include financial constraints, concerns about family welfare, and anxiety about having to change the roles and responsibilities of family members [23].

### 5.7. The Child’s Activity

Similar to the findings of the present research, studies have shown increased depression among parents in cases of functional impairment and decreased performance of the child [60,61,62]. The disease symptoms and the tiring treatment with risks of many complications contribute to limiting the child’s activity, and thus to increasing stress and mental health disorders in parents [62].

### 5.8. Age of the Child

The study by Al Qadire et al. confirms the results of the present research on the relationship between the age of the child and the severity of the parents’ disorders. The illness of a younger child causes parents more suffering, stress, and anxiety [63,64]. The reason for this is that younger children require more intensive care than older children [64]. On the other hand, the study by Eyigor et al. reported a negative correlation between the age of the child and the severity of mental disorders in the parents, explaining that older children are able to understand what illness is and therefore take on the role of a patient. They are also more independent than younger children, making it harder to control their perception of the illness. Awareness of illness in older children and their resultant low mood causes deterioration of their carers’ mental health [65].

### 5.9. Co-Occurrence of Mental Disorders

The present study and the available literature indicate that anxiety affects the severity of depression, and vice versa [63]. Symptoms of depression, anxiety, and PTSD can coexist, and their co-occurrence correlates strongly with more severe forms of depression [40]. The authors’ own research did not show a correlation between the type of cancer in the child and the severity of depression in the parent, but the available literature indicates an increased likelihood of greater distress after receiving a diagnosis of solid tumours compared with haematological cancers. This is most likely due to the fact that even in the case of advanced stage haematological cancers, treatment is much more effective than for solid tumours and 10-year survival rates are higher [66,67,68,69]. In addition, single parents present a higher risk of developing depression than parents raising a child together [35].

### 5.10. Social Support

An important aspect in these circumstances is social support. Parents of affected children often communicate with each other by joining various support groups or foundations where the experiences of other parents are similar to their own, which provides them with support and contributes to reducing the severity of mental disorders and improving quality of life [35,70,71]. Lack of social support is a major predictor of depression [66]. Studies show that support from friends has a positive effect on coping with and overcoming adversity [72].

### 5.11. A Task for Clinicians

Properly communication of information regarding the child’s condition is an extremely important skill for clinicians, and should be carried out in a way that is understandable to the parent and adjusted to their level of knowledge.

In conclusion, identifying depression in parents of children with cancer and providing them with appropriate support significantly affects their attitudes towards the illness, and can improve quality of life for the whole family. Increased severity of anxiety and depression contribute to a worsening course of the child’s illness, less effective treatment, and increased personal, social, and medical costs.

The present study proposes focus on this topic, to prevent and effectively treat depression and anxiety in carers of children with cancer. The task for the health service is to assess risks of depression and anxiety and to assess mental health and coping mechanisms among parents of children with cancer [73]. This will allow timely prevention of the disorders and prompt provision of psychological care or treatment, in order to improve quality of life for the whole family [63]. Parents should be included in a psychological intervention programme, based on family cognitive and behavioural therapies [73], and should receive social and even financial support [25].

## 6. Conclusions

Receiving a diagnosis of cancer in a child contributes to the incidence of depression and anxiety disorders among carers.

A major finding of this study is that carers’ anxiety and depression levels are strongly associated with the child’s age and the child’s performance. The younger the child and the lower the levels of performance, the greater the severity of depressive disorders.

## 7. Limitations of the Study

The effect estimates in the study were based on observational studies. A potential limitation of the study is the lack of comparison between anxiety and depression levels among caregivers depending on their marital status or level of education. A low level of education and a marital status of divorcee or widow/er could potentially exacerbate a parent’s levels of anxiety and depression. However, caregivers who participated in the survey did not consent to the provision of such sensitive personal data.

## 8. Implications

The present study provides important information about depressive and anxiety disorders among parents of children with cancer. The findings have implications for public health professionals, the government, and future researchers.

We recommend that public health providers take an interest in mental health and coping mechanisms among parents of children with cancer, in order to prevent or treat anxiety and depression and to improve carers’ quality of life.

We would like to suggest to the government that there is a need to create certain educational projects, which would aim to educate society about mental disorders, and thereby help people suffering from anxiety and depression by increasing the available levels of social support. A psychological intervention programme should be created, based on family cognitive and behavioural therapies. Moreover, the parents of children with cancer should receive financial support.

We suggest that future researchers use Online Photovoice (OPV) to conduct further study on the topic of mental disorders among parents and other family members. OPV gives opportunities to participants to express their own experience with as little interference as possible, if any, compared with traditional quantitative methods.

## Figures and Tables

**Table 1 jcm-11-05670-t001:** General statistics of the children (age, diagnosis, disease duration).

Age [Years]	*n*	Mean	SD	Median	Min.	Max.	Q1	Q3	IQR
	100 *	8.87	4.9	8	1.17	18	5	13	8
Diagnosis	Type of Cancer			*n*		**%**			
	Leukaemia			58		57.43%			
	Lymphoma			5		4.95%			
	Sarcoma			9		8.91%			
	Nervous system tumour			18		17.82%			
	Other types of cancer			10		9.90%			
	No answer			1		0.99%			
Disease duration	Time			*n*		%			
	Up to 6 months			34		33.66%			
	7–12 months			33		32.67%			
	13–24 months			12		11.88%			
	Over 24 months			21		20.79%			
	No answer			1		0.99%			

* One parent did not fulfilled the information.

**Table 2 jcm-11-05670-t002:** Zung Self-Rating Depression Scale.

Number of Points	Interpretation	*n*	%
<50	No depression	90	89.11%
50–59	Mild depression	10	9.90%
60–69	Moderate depression	1	0.99%
70 and more	Severe depression	0	0.00%

**Table 3 jcm-11-05670-t003:** HADS—anxiety and depression.

Anxiety—Number of Points	Interpretation	*n*	%
0–7	No disorders	36	35.64%
8–10	Borderline state	23	22.77%
11–21	Marked disorders	42	41.58%
Depression— Number of Points	Interpretation	*n*	%
0–7	No disorders	52	51.49%
8–10	Borderline state	26	25.74%
11–21	Marked disorders	23	22.77%

**Table 4 jcm-11-05670-t004:** Empirical correlations between anxiety levels and depression levels among carers.

Parameter	Correlation with Anxiety (HADS)			
	Correlation Coefficient	*p*	Relationship Direction	Relationship Strength
HADS—Depression	0.863	<0.001	positive	strong
Zung scale	0.833	<0.001	positive	strong

**Table 5 jcm-11-05670-t005:** Children’s performance according to the Karnofsky scale.

Karnofsky Performance Scale								
*n* *	Mean	SD	Median	Min.	Max.	Q1	Q3	IQR
100	68.8	20.71	70	10	100	57.5	80	22.5

* One parent did not complete the Karnofsky scale questionnaire.

**Table 6 jcm-11-05670-t006:** Carers’ anxiety and depression levels and children’s performance.

Parameter	Correlation with Karnofsky Scale			
	Correlation Coefficient	*p*	Relationship Direction	Relationship Strength
HADS—Anxiety	−0.33	0.001	negative	weak
HADS—Depression	−0.311	0.002	negative	weak
Zung scale	−0.247	0.013	negative	very weak

**Table 7 jcm-11-05670-t007:** Daily activity of carers of children with cancer.

Statement	Yes		No		I Don’t Know		No Answer	
	*n*	%	*n*	%	N	%	*n*	%
I go out to meet my family and friends less often	87	86.14%	11	10.89%	3	2.97%	0	0.00%
I show less interest in other people’s problems	13	12.87%	63	62.38%	25	24.75%	0	0.00%
Other people often irritate me	22	21.78%	61	60.40%	18	17.82%	0	0.00%
I show less emotion	10	9.90%	76	75.25%	15	14.85%	0	0.00%
I have less contact with people	43	42.57%	48	47.52%	10	9.90%	0	0.00%
My relationship with my immediate family and friends has deteriorated	29	28.71%	62	61.39%	10	9.90%	0	0.00%
I pay less attention to my appearance	35	34.65%	30	29.70%	35	34.65%	1	0.99%
I cannot concentrate for a long time	32	31.68%	40	39.60%	28	27.72%	1	0.99%
I work less or no hours	81	80.20%	16	15.84%	2	1.98%	2	1.98%
I am less efficient at work	36	35.64%	17	16.83%	43	42.57%	5	4.95%
I leave the house less often for entertainment purposes	88	87.13%	7	6.93%	4	3.96%	2	1.98%
I spend less time on my passions and hobbies	79	78.22%	14	13.86%	7	6.93%	1	0.99%
I have completely given up active leisure	62	61.39%	30	29.70%	8	7.92%	1	0.99%
I still enjoy an active life	49	48.51%	13	12.87%	38	37.62%	1	0.99%

**Table 8 jcm-11-05670-t008:** Type of complications after chemotherapy.

Complications after Chemotherapy	*n* *	% *
Decreased appetite	89	88.12%
Weakness	83	82.18%
Dizziness	37	36.63%
Heart palpitations	13	12.87%
Easy fatigability	42	41.58%
Pale skin	67	66.34%
Nose or gum bleeding	13	12.87%
Oral lesions	42	41.58%
Sialorrhea	2	1.98%
Pain in the oral cavity	52	51.49%
Difficulty swallowing	23	22.77%
Swelling of the face or neck	17	16.83%
Diarrhoea	48	47.52%
Nausea and vomiting	77	76.24%
Fever	47	46.53%
Abdominal pain	63	62.38%
Hair loss	78	77.23%
Skin discolouration	32	31.68%
Cough	7	6.93%
Hemoptysis	2	1.98%
Difficulty breathing	8	7.92%
Other	13	12.87%

* Percentages do not sum to 100% because that was a multiple-choice question.

**Table 9 jcm-11-05670-t009:** Mean severity of selected complications in children after chemotherapy.

Complication	*n* *	Mean	SD	Median	Min.	Max.	Q1	Q3	IQR
Decreased appetite	100	5.16	2.29	5	0	10	4	6	2
Weakness	100	4.59	2.28	4	0	10	3	6	3
Headache	100	2.13	2.51	1	0	9	0	4	4
Dizziness	100	1.81	2.2	1	0	9	0	3	3
Pain in the oral cavity	100	3.09	2.98	3	0	10	0	5.25	5.25
Diarrhoea	99	2.54	2.35	2	0	9	0	4	4
Constipation	99	2.23	2.55	2	0	10	0	4	4
Abdominal pain	100	3.61	2.57	4	0	10	2	5	3
Nausea and vomiting	100	4.39	2.67	4	0	10	2.75	6	3.25
Fever	100	2.36	2.15	2	0	9	0	4	4

* Not all parents assessed the severity of all complications.

**Table 10 jcm-11-05670-t010:** The association between children’s age and the levels of depression and anxiety in their carers.

Parameter	Correlation with the Child’s Age			
	Correlation Coefficient	*p*	Relationship Direction	Relationship Strength
HADS—Anxiety	−0.121	0.23	-	-
HADS—Depression	−0.165	0.101	-	-
Zung scale	−0.255	0.011	negative	very weak

**Table 11 jcm-11-05670-t011:** The association between the children’s disease duration and carers’ anxiety and depression.

Parameter	Correlation with the Disease Duration			
	Correlation Coefficient	*p*	Relationship Direction	Relationship Strength
HADS—Anxiety	0.035	0.731	-	-
HADS—Depression	0.027	0.791	-	-
Zung scale	0.084	0.406	-	-

## Data Availability

Not applicable.

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
