# Peer review of "Depressive and Anxiety Disorders of Parents of Children with Cancer"

_jcm, 2022, doi:10.3390/jcm11195670_

Round 1
Reviewer 1 Report
I have enjoyed reading the paper and am looking forward to seeing the paper published. You could increase the effect of your paper with some more recent studies suggested below or any other studies.
Title of the manuscript: “Depressive and anxiety disorders of parents of children with cancer”
Dear researcher(s), you are addressing an important and meaningful gap. Your paper is well-written and it has some important results, and if you edit your paper it can be much more effective. Here some humble suggestions to improve the paper, I would do the following to strengthen the paper. I have enjoyed reading the paper and am looking forward to seeing the paper published. You could increase the effect of your paper with some more recent studies suggested below or any other studies.
Main points:
1. Title: Good, brief and and appropriate
2. Abstract and keywords clear and comprehensive: This is good but with minor corrections in yellow font (line 16 and 25 -26). It is advisable to start a sentence with word and not numbers i.e. 101 caretakers should be One hundred and one caretakers (line 16). The aim of the study should be mentioned earlier after background rather than after conclusion (line 25 – 26).
3. Overall language:
- The language should be proof read. You could use an active language for your future papers throughout the paper since an active language seems to be more effective. And more and more researchers go with an active language. However, you do not have to change for this paper- just a suggestion for your future work and I know some journals asking for a passive language.
4. Length of paragraph: The paragraph needs to be revisited. It is advised that paper should be paragraphed well.
a. you can check the paper and make sure every paragraph is not more than 5 sentences. The best is to stick with 3 to 5 sentences.
b. And you can add some subtitles to be more organized and short sections. Please see the suggested papers above. I would suggest you to look at the suggested studies and use more subtitles to make the paper more organized.
5. Introduction: The introduction should be written based on the aim of the study. Some corrections in yellow font (lines 34, 35, 36) noticed. Line 34 [2, 3] should have full stop at the end. Line 35 acute lymphoblastic leukaemia should be abbreviated (ALL) before it is presented in line 36.
6. Thoroughness of the literature review: can support with some recent studies yet do not have to. This will increase the effect of the paper and the journal.
7. Clarity of the description of the Theoretical Framework (TF): Not applicable, please see suggested studies as an example
8. Research design: The type of survey conducted was not mentioned which is a major omission on the paper. Advised to re-visit this section. 101 should be first written in words at the beginning of the sentence.
9. Clearly providing research questions and/or purpose: Not mentioned in study.
10. Choice of research method: Needs to be mentioned cleared.
11. Appropriateness of procedures chosen for data collection and analysis: well-written
12. Relevance of data obtained in view of the purpose of the research: well-written
13. Discussion of the results and their significance: Advised to proof read the section and organize based on objectives of the study.
14. Soundness of conclusions in relation to data presented: Advised to re-write this section.
15. Limitation: Advised to improve the context here.
16. Implication: Advised as below please.
a. you can increase the effect of your paper by constructing a new section entitled “implication” for clear and brief suggestions in at least two or three of the following most important to you mental health, education, research, administrators, services, etc.: see suggested papers for implications for specific sections
b. I would strongly suggest you to call future researchers to use Online Photovoice (OPV) to conduct research on the same or similar topics. The researchers can use OPV, as one of the most recent and effective innovative qualitative research methods. OPV gives opportunities to the participants to express their own experience with as little manipulation as possible if at all, compared to traditional quantitative methods. As researchers one of our responsibilities is to inform others about recent and effective methods, which will increase the effect of your paper and the journal. Future researchers can conduct only qualitative or mixed method to see if OPV. And educators/trainers etc. also can use OPV for experiential activities to increase group and organizational synergy. Please see suggested papers if you wish to do so.
Armiya’u, A. Y., Yildirim, M., Muhammad, A., Tanhan, A., & Young, J. S. (2022). Mental health facilitators and barriers during covid-19 in Nigeria. Journal of Asian and African Studies. https://doi.org/10.1177/00219096221111354
https://www.researchgate.net/publication/361990976_Mental_Health_Facilitators_and_Barriers_during_Covid-19_in_Nigeria
DoyumÄŸaç, İ., Tanhan, A., & Kıymaz, M. S., (2021). Understanding the most important
facilitators and barriers for online education during COVID-19 through online photovoice methodology. International Journal of Higher Education, 10(1), 166-190. https://doi.org/10.5430/ijhe.v10n1p166
https://scholar.google.com/scholar?hl=en&as_sdt=0%2C5&q=Understanding+the+most+important++facilitators+and+barriers+for+online+education+during+COVID-19+through+online+photovoice+methodology&btnG=
Tanhan, A., & Strack, R. W. (2020). Online photovoice to explore and advocate for Muslim
biopsychosocial spiritual wellbeing and issues: Ecological systems theory and ally development. Current Psychology, 39(6), 2010-2025. https://doi.org/10.1007/s12144-020-00692-6
https://scholar.google.com/scholar?hl=en&as_sdt=0%2C5&q=Online+photovoice+to+explore+and+advocate+for+Muslim++biopsychosocial+spiritual+wellbeing+and+issues%3A+Ecological+systems+theory+and+ally+development&btnG=
17. Figure/tables: Good and clear
18. References:
- You could increase the effect of your paper with some more recent studies
- Please use the following link to include all available doi numbers https://doi.crossref.org/simpleTextQuery simply include your reference one or more than one at a time and submit it. Then you should get all doi numbers if a manuscript has it.
I have enjoyed reading your paper and learned a lot- thanks for your contribution to social sciences. You are addressing an important and meaningful gap. Your paper has some important results, and if you edit your paper based on all or some of the humble suggestions above, it can be much more effective. I am looking forward to seeing the paper published. You could increase the effect of your paper with some more recent studies suggested above or any other studies and not using the suggested ones.
Reviewer 2 Report
The reviewed paper is settled in the clinical context and deals with the psychological well-being of parents whose children suffer from cancer. After a short introduction into this topic, the authors present results of a study with n=101 caretakers of children suffering from cancer. After the presentation of descriptive sample statistics, the authors examine the possible link between parents’ level of anxiety and depression and children’s health performance. Indeed, they found several significant correlations between parents’ psychological well-being and children’s characteristics (e.g., “the older the child, the lower the severity of the disorder”, page 4, line 167). The authors concluded that children’s cancer disease affects the psychological well-being of the whole family and, thus, decreases parents’ quality of life.
I really enjoyed reading this article and thank the authors for the opportunity to review their interesting paper. In my opinion, the paper is well written and structured. What I liked most about the article is the high quality of the presented results. I (nearly) missed nothing in the detailed Tables, all necessary statistical parameters are reported. The written English language is very good and the reader is introduced smoothly into this topic.
However, I missed the “added value” of this paper. It sounds trivial to me that parents’ psychological well-being is connected to their children’s well-being. Moreover, the aim of the study was “to assess the level of anxiety and depression of caretakers of children with cancer and to present the influence of the child's age, the number of complications connected with chemotherapy and the performance of the child on the severity of parents' mental health disorders” (page 2, line 54 and following lines). From a statistical point of view, I would have expected regression analyses with the level of anxiety and depression of caretakers as outcome variables. Children’s characteristics like age, health status etc. are used as predictor variables. Unfortunately, the authors present correlation coefficients only what limits the interpretation of the results. In this sense, causality cannot be inferred from presented correlations. Thus, I would urge the authors to add multiple linear regression analyses in their method and result sections.
Besides these two major issues, I detected some smaller errors and miss-spellings (note, this list in not complete). Hence, I have some comments/suggestions that I hope will help the authors to further develop this line of work:
- Abstract (page 1, line 16): The abbreviation PTSD has not yet been introduced for the reader. Please write “post-traumatic stress disorder (PTSD)” at this point.
- Page 3, line 99: Please introduce a new chapter at this point with the heading “Statistics” for the statistical tests and parameters used in this paper. It is not appropriate to mention these in the chapter “2. Materials and Methods”.
- Page 3, line 99: The authors forgot to mention the first and third quartile which are later presented in tables (e.g., Table 1, Q1 and Q3). Why do the authors not report the interquartile range (IQR) since the IQR is the “standard deviation” (SD) in the non-parametric case (i.e., parametric case: mean (M) and standard deviation (SD); non-parametric case: Median (MD) and interquartile range (IQR)).
- Page 3, line 112: “[…], a significance level of p=0.05 was adopted”. Here, the authors mixed up the significance level Alpha (α) and the p-value which are both not equal. Please write either “[…], a significance level of α=0.05 was adopted” or “[…], p-values ≤0.05 are regarded as statisticially significant”. Please note the difference between “=” and “≤”.
- Page 3, line 133 and page 5, line 177 (Table 4): “Using Spearman’s correlation coefficient, a statistically significant effect of […]”. As I mentioned above, this sentence is wrong. A (non-parametric) Spearman’s correlation detects associations (and not effects) between two variables. Hence, there is no direction between both variables implied, no causality. Also, the authors write in the table heading of Table 4 “The effect of anxiety levels on the prevalence of depression among caretakers”, implying that anxiety “effects” depression but this is wrong. Correlation coefficients are only associations (x -> y, but also y -> x), and, in the worst case, can be influenced by confounders, resulting in spurious correlations, an association without any meaning. Please correct this and use the term “effect” only in the case of regression analyses. For example, the authors can write “Empirical associations/correlations between anxiety levels and depression levels among caretakers”.
- Page 5, line 181 (Table 7): Please left-allign table heading
- Page 6, line 184 (Table 9) Please left-allign table heading
- Page 7, line 186 (Table 10): Please left-allign table heading
- Page 7, line 186 (Table 10): not effect but rather association/relationship
- Page 7, line 187 (Table 11): Please left-allign table heading
- Page 7, line 187 (Table 11): not effect but rather association/relationship
- Page 8, line 204: not „[…] faith and trust. These stages […]“ but rather […] faith and trust. These stages […]“. Please delete the unnecessary space between the dot and the word „These“.
- Page 8, line 208: The abbreviation PTSD has been already introduced before (see page 2, line 49). Please introduce this abbreviation only once (1) in the abstract (stand-alone text) and (2) in the article.
- Page 8, line 248: not „cooccurrence“ but rather „co-occurrence“. Please correct this.
Round 2
Reviewer 2 Report
I thank the authors for submitting a revised version of their manuscript entitled “Depressive and anxiety disorders of parents of children with cancer”.
The authors did a great job, improved their manuscript according to my suggestions, and responded to nearly all my questions adequately. In my opinion, the revised manuscript increased a lot in comparison to the first version. I have one minor point left:
· The authors did not follow my earlier suggestion to add multiple linear regression analyses to the result section and maintained their correlation analyses only. On the one hand, this is fine for me, because they changed the word “effect” to “associations” in the entire manuscript. Now, the reader is not misdirected anymore. However, on the other hand, I do not follow the final conclusion of this manuscript: “The factors significantly affecting the severity of depressive and anxiety disorders are: the child’s age and the levels of performance” (page 10, line 318). This conclusion does not match the presented results. This sentence implies that child’s age and child performance are predictor variables for caretakers’ anxiety and depression levels meaning that there is a direction (i.e. causality) between these variables (i.e., caretakers’ high anxiety and depression levels are caused by child’s low age and child’s low performance). Although that would be reasonable on a theoretical level, the statistical analyses (i.e., correlation analyses) are not quite adequate for this argumentation.
In other words, the authors present in Table 6 (page 5, line 198) significant correlations between caretakers’ anxiety and depression levels and child performance. However, this association does not imply that a low child performance causes high anxiety and depression levels in caretakers. The opposite direction is also possible (caretakers get anxious and depressive for other reasons what reduces child’s performance). The same is true for caretakers’ anxiety and depression levels and child’s age. Author’s current conclusions (i.e., page 10, line 318) would imply that a low child’s age (when getting the cancer diagnosis) is the cause for caretakers’ high anxiety and depression levels. However, correlation analyses do not account for causality.
Hence, I would urge the authors to change this sentence in the conclusion section from “The factors significantly affecting the severity of depressive and anxiety disorders are: the child’s age and the levels of performance” (page 10, line 318) to “As a major result, we found in this study that caretakers’ anxiety and depression levels are strongly associated with child’s age and child’s performance” (or similar formulations). The same is true for the conclusion in the abstract.
Author Response
Dear reviewer, Thank you for your opinion.We are so thankful that you think our work has improved and has increased a lot in comparison to the first version.
We are immensely grateful for your advice.
We understood our mistakes and learned for the future that correlation analyzes do not account for causality.
Thanks to this, in our future studies, we will definitely focus on performing multiple linear regression analyzes to make the work more valuable.
In accordance with your request, we have changed the sentences in the sections: conclusions and in the abstract.